# Avian Influenza: Strategies to Manage an Outbreak

**DOI:** 10.3390/pathogens12040610

**Published:** 2023-04-17

**Authors:** Alison Simancas-Racines, Santiago Cadena-Ullauri, Patricia Guevara-Ramírez, Ana Karina Zambrano, Daniel Simancas-Racines

**Affiliations:** 1Facultad de Ciencias Agropecuarias y Recursos Naturales, Carrera de Medicina Veterinaria Universidad Técnica de Cotopaxi, Latacunga 050108, Ecuador; 2Centro de Investigación Genética y Genómica, Facultad de Ciencias de la Salud Eugenio Espejo, Universidad UTE, Quito 170129, Ecuador; 3Centro de Investigación de Salud Pública y Epidemiología Clínica (CISPEC), Universidad UTE, Quito 170129, Ecuador

**Keywords:** virus, infectious disease, diagnosis, control, biosecurity, poultry

## Abstract

Avian influenza (AI) is a contagious disease among the poultry population with high avian mortality, which generates significant economic losses and elevated costs for disease control and outbreak eradication. AI is caused by an RNA virus part of the *Orthomyxoviridae* family; however, only *Influenzavirus A* is capable of infecting birds. AI pathogenicity is based on the lethality, signs, and molecular characteristics of the virus. Low pathogenic avian influenza (LPAI) virus has a low mortality rate and ability to infect, whereas the highly pathogenic avian influenza (HPAI) virus can cross respiratory and intestinal barriers, diffuse to the blood, damage all tissues of the bird, and has a high mortality rate. Nowadays, avian influenza is a global public health concern due to its zoonotic potential. Wild waterfowl is the natural reservoir of AI viruses, and the oral–fecal path is the main transmission route between birds. Similarly, transmission to other species generally occurs after virus circulation in densely populated infected avian species, indicating that AI viruses can adapt to promote the spread. Moreover, HPAI is a notifiable animal disease; therefore, all countries must report infections to the health authorities. Regarding laboratory diagnoses, the presence of influenza virus type A can be identified by agar gel immunodiffusion (AGID), enzyme immunoassay (EIA), immunofluorescence assays, and enzyme-linked immunoadsorption assay (ELISAs). Furthermore, reverse transcription polymerase chain reaction is used for viral RNA detection and is considered the gold standard for the management of suspect and confirmed cases of AI. If there is suspicion of a case, epidemiological surveillance protocols must be initiated until a definitive diagnosis is obtained. Moreover, if there is a confirmed case, containment actions should be prompt and strict precautions must be taken when handling infected poultry cases or infected materials. The containment measures for confirmed cases include the sanitary slaughter of infected poultry using methods such as environment saturation with CO_2_, carbon dioxide foam, and cervical dislocation. For disposal, burial, and incineration, protocols should be followed. Lastly, disinfection of affected poultry farms must be carried out. The present review aims to provide an overview of the avian influenza virus, strategies for its management, the challenges an outbreak can generate, and recommendations for informed decision making.

## 1. Introduction

Avian influenza (AI), also known as “bird flu” is a contagious disease among the poultry population with high avian mortality, causing a decrease in production, generating economic losses, and limiting the circulation of birds, its products, and subproducts. Moreover, disease control and outbreak eradication represent high costs [1].

AI is an infection caused by influenza A virus, whose nomenclature has been assigned by the World Health Organization (WHO), the World Organization for Animal Health (WOAH), and the Food and Agriculture Organization of the United Nations (FAO) [2], according to their transmembrane hemagglutinin (HA) and neuraminidase (NA) glycoproteins [3]. AI pathogenicity is based on the lethality, signs, and molecular characteristics of the virus and is classified as being high or low pathogenic. Low pathogenic avian influenza (LPAI) is a mild disease, often unnoticed or asymptomatic. In contrast, highly pathogenic avian influenza (HPAI) causes severe systemic affections with high morbidity and mortality rates [4].

Until now, it has been not clear how the geographic distribution of avian influenza occurred according to its pathogenic lineages. However, from 1880 to 1959, AI spread in birds throughout Europe, Asia, Africa, and North and South America [5]. The LPAI virus was first identified in poultry in Germany in 1949 as H10N7. Furthermore, the first LPAI virus in wild waterfowl and pelagic seabirds was discovered in 1972 [6].

The first HPAI virus was identified in 1880 in Northern Italy in domestic birds [5]. Later, in 1996, the H5N1 virus was identified in waterfowl in China [7]; moreover, by 2003 and 2005, the virus diversified its genetic lineages and crossed borders, causing widespread outbreaks in poultry in Asia, Africa, Europe, the Middle East, and North America [8]. Due to this variability, between 2018 and 2020, outbreaks of H5N6 and H5N8 subtypes emerged and predominated worldwide [7]. A new HPAI outbreak of a novel H5N1 subtype was reported in Asia, Africa, Europe, and the Middle East by the end of 2021 and in Canada and the USA by 2022 [7].

Moreover, AI is considered a notifiable animal disease by the WOAH and FAO; therefore, all countries must report to the health authorities infections by HPAI and low pathogenic H5 and H7 subtypes due to their ability to mutate to HPAI [9].

HPAI has caused the death and mass depopulation of approximately 316 million poultry between 2005 and 2021, affecting more than 50 countries, especially in 2021, 2020, and 2016 according to reports generated by animal health control agencies [10]. In 2023, 37 outbreaks in poultry and 120 outbreaks in wild birds were reported in Europe, Asia, North America, and Latin America [11]. The outbreaks resulted in about 2.5 million poultry culled worldwide, with the H5N1 subtype predominating in all cases [10,11,12].

Avian influenza is a global public health concern due to its zoonotic potential [13,14,15]. Moreover, there are reports of outbreaks in free-living wild and captive mammals [12,16]. The most recognized zoonotic AI virus is the H5N1 subtype, which first appeared in humans in 1997 in Hong Kong and then re-emerged in mainland China in 2003 [17]. Human cases have been reported globally with subtypes H5N1 (870 cases and 430 deaths); H7N9 (1500 cases and 600 deaths); H5N6 (80 cases and 30 deaths); H9N2 (80 cases and 2 deaths); and sporadic reported cases of subtypes H3N8, H7N4, H7N7, and H10N3 [15,18,19,20]. From 2003 to 2022, 868 human cases of influenza A (H5N1) infection causing 457 deaths were reported to PAHO/WHO in 21 different countries. In 2022 alone, several human cases of avian influenza have been reported: the United Kingdom, Canada, and Spain reported cases of HPAI (H5N1) identified in poultry farm operators [21], and Asian countries reported infections with other subtypes, namely H5N6, H3N8, and H9N2 [22], of which there is already one fatal case [23].

The present review aims to provide an overview of the avian influenza virus, strategies for its management, the challenges an outbreak can generate, and recommendations for informed decision making.

## 2. Avian Influenza Virus

### 2.1. Etiology

The influenza virus is an RNA virus part of the *Orthomyxoviridae* family with seven genera, namely *Influenzavirus A*, *Influenzavirus B*, *Influenzavirus C*, *Influenzavirus D*, *Thogotovirus*, and *Isavirus* and *Quarajavirus* [24,25]; moreover, *Influenzavirus A* has been identified in a wide range of hosts with the highest genetic variability and is the only one capable of infecting birds [26,27]. Moreover, due to the segmented nature of the viral genome, new strains can emerge through genetic reassortment and antigenic drift, further increasing the difficulty in its control and prevention [28].

The AI virus subtypes depend on the antigen present on the surface of the influenza A virus; there are 16 hemagglutinin subtypes and 9 neuraminidase subtypes [29,30]. However, recent scientific studies reported new HA subtypes (18 in total) and NA (11 in total), which were isolated in bats [31].

Additionally, there are two specific lineages of the HA subtype that phylogenetically divide into the Eurasian and North American lineages; these lineages have progressively evolved independently due to limited intercontinental contact between avian populations [32].

### 2.2. Pathogenicity and Virulence

AI viruses generally cause gastrointestinal disturbances in birds with minimal clinical signs and are classified as LPAI viruses. LPAI virus subtypes H5 and H7 circulate naturally in domestic birds but can evolve and become highly pathogenic [33].

Pathogenicity results from the accumulation of multiple basic amino acids at the HA cleavage site (termed the polybasic cleavage site or polybasic motif), allowing the HA molecule to develop outside the gastrointestinal tract and establish a systemic infection, causing an outbreak of HPAI that is characterized by rapid disease onset and progression associated with high mortality rates [4,32,34]. Moreover, some H5 and H7 viruses of low and high pathogenicity show virulence in mammals, and the highly pathogenic viruses can cause systemic infection in animal models [32,35,36].

#### 2.2.1. Low Pathogenic Avian Influenza Virus (LPAI)

LPAI has a low mortality rate and ability to infect, causing little to no disease in birds, because they can only replicate in tracheal tissues and the small intestine [24]. However, the H5/H7 subtypes of low pathogenicity (common in poultry and wild waterfowl) [37] can mutate by insertion and recombination processes in the proteolytic cleavage site of HA [6] until becoming HPAI viruses [38].

#### 2.2.2. Highly Pathogenic Avian Influenza Virus (HPAI)

The HPAI virus can cross respiratory and intestinal barriers, diffuse to the blood, and damage all tissues of the bird [37]. HPAI refers to strains with an “intravenous pathogenicity index” (IVPI) greater than 1.2 or a mortality rate equal to or higher than 75% of the total number of poultry over a period of 10 days [39].

The HPAI pathogenic strains of avian influenza belong to the H5 and H7 subtypes, with bird mortality that exceeds 90–100% during the 48 h after disease onset [39,40].

To date, subtypes H5 and H7 have been recognized as HPAI viruses capable of generating acute and considerable diseases in chickens, turkeys, and other economically significant birds. Moreover, H9 has been included as another subtype with pandemic risk because their high mutability could favor the evolution of viruses that allow sustained transmission in the human species, and H9 can cause zoonotic infections [37].

### 2.3. Transmission Mechanisms

#### Bird-to-Bird Transmission

Wild waterfowl are natural reservoirs of the AI virus and play a role in spreading through their long-distance migratory routes [41], infecting land birds and domesticated waterfowl via contaminated water sources or food [14]. However, the oral–fecal path is the main transmission route between birds due to the high viral levels in the fecal matter of infected birds, and it can be transmissible for approximately 21 days [17,25].

Chatziprodromidou et al. described proximity to water as a significant risk factor for virus transmission because there may be a close interaction between migratory birds and commercial poultry activities, increasing disease transmission [9].

AI virus can also be transmitted through secretions and body fluids, such as saliva, mucus, and urine [5]. In the production systems, these fluids and feces contaminate the clothing and footwear of operators, cages, implements, and mechanical equipment for egg collection, among others. This route has been considered the principal vehicle for disease dissemination within flocks [1], making commercial poultry responsible for epidemics registered worldwide [42].

### 2.4. Interspecies Transmission

#### 2.4.1. Transmission to Mammals

Direct contact is the main route of transmission because it has not been demonstrated that the virus can effectively infect mammals through aerosols [43]. Transmission to other species generally occurs after virus circulation in densely populated infected avian species, indicating that AI viruses can adapt to promote the spread [14]. For effective transmission and replication in mammals, the virus must evolve and mutate until it reaches compatibility with the new host environment; this is known as viral reassortment, which has been responsible for the appearance of almost all pandemic viruses in the past [44,45,46].

Infections with avian influenza virus have been reported in cats, mice, and pigs with AI subtype H5N6 [47,48,49]; in canines with subtype H3N8 [50]; and in tigers and leopards with subtype H1N1 [51,52]. All of them have been epidemiologically related to avian influenza outbreaks. Furthermore, avian influenza subtypes have been isolated in ferrets and laboratory animals to evaluate their pathogenicity [53,54].

#### 2.4.2. Zoonotic Transmission

Avian influenza viruses have demonstrated the capacity to cross the barrier between species for multifactorial reasons that have favored transmission. Certain mammals, such as bats [55], pigs [56], cats, dogs, horses, ferrets, sea lions, and bats [57], can act as reservoirs, which allow genetic mixing between viruses that intend to infect humans and birds [39]. Moreover, host susceptibility, exposure level to infected birds, viral mutations, and favorable environmental conditions form an ideal scenario for the zoonotic transmission of the avian influenza virus [58].

The main route of transmission between birds and humans is direct contact with the feces or secretions of infected animals and exposure to contaminated or virus-infected environments (Figure 1) [59,60]. There is no evidence of human-to-human infection [5,14]. People within the poultry production chain (from farm to table) are at a higher infection risk than the general population due to prolonged exposure to the infectious agent [4].

### 2.5. Virus Reservoirs

The avian influenza virus ecological niche or natural reservoir are waterfowl belonging to *Anseriformes* (waterfowl, ducks, geese, and swans) and *Charadriiformes* (gulls and shorebirds), which include more than 100 species of wild birds belonging to about 25 different families, indicating the global distribution of the virus in free-living waterfowl [34,61].

Subtype AI viruses (H5, H7, H6, and H9) can be found in both waterfowl and poultry [62], and several papers have described avian influenza viruses identified in different mammals. For instance, the AI subtype (H3, H7) in equines, the AI subtype (H1, H3) in swine, and the AI subtype in aquatic mammals (H10, H4, H7, and H13) originate from the genetics inherent to viruses naturally found in wild waterfowl (H1–H16) [25,63].

### 2.6. Virus Survival

Studies have found that the virus is more resistant to low temperatures (below 28 °C) [1,64]. The avian influenza virus can survive for up to 200 days in the body fluids of infected birds, four days in feces at animal body temperature, 35 days in feces at temperatures below 4 °C, and about five weeks in the environment of the infected poultry house [1,64]. The virus can survive in carcasses, meat, and eggs (especially at low temperatures); therefore, upon suspicion or confirmation of positive cases of avian influenza, the products generated should be eliminated [64].

## 3. Avian Influenza in Poultry

### 3.1. Clinical Diagnosis

Clinical signs of avian influenza vary according to the strain or subtype involved in the outbreak, level of immunity, age, and type of bird presenting with the infection [65]. The signs range from mild upper respiratory problems affecting egg production [66] to fatal systemic disease [67]. However, infected birds may die without indications of avian influenza [40].

#### 3.1.1. Highly Pathogenic Avian Influenza (HPAI)

In infected birds with a weak immune system, sudden death can occur without evident clinical signs, or manifest as a type of respiratory disease with signs such as ocular and nasal secretions, cough, snorting, dyspnea, nasal sinuses swelling, apathy, reduction in vocalization, marked reduction in feed and water intake, cyanosis, skin without feathers, lack of coordination and nervous signs, and digestive problems that are evidenced by diarrhea [5,10]. In laying hens, there is an evident deterioration in the quality and quantity of egg production [5,37].

#### 3.1.2. Low Pathogenic Avian Influenza (LPAI)

LPAI viruses generally cause only a mild or subclinical disease; however, if there are comorbidities in the animals or unfavorable environmental conditions, the clinical picture may be similar to that of HPAI [5,37].

### 3.2. Laboratory Diagnosis

#### 3.2.1. Sampling

It is suggested to take trachea, lung, and intestine samples from a minimum of five sick birds and report the results. Moreover, cloacal and tracheal swabs from 10 to 30 healthy birds and at least one gram of fecal matter and acute sera from 10 birds should be collected [40]. Samples should be placed at ≤−20 °C (for no more than seven days) or ≤−70 °C and transported promptly. Sample freezing and thawing should be avoided [68]. Clinical samples of human and avian origin should not be tested at the same site to avoid genetic recombination or the reassortment of animal and human strains [40].

#### 3.2.2. Viral Isolation

Inoculation of specific pathogen-free (SPF) embryonated chicken eggs has been the method of choice for cell cultures of avian influenza virus [69]. The samples are inoculated into the allantoic cavity of 9- to 11-day-old embryonated hen eggs and incubated at 37 °C for approximately 2 to 7 days. Eggs are tested for hemagglutinating activity at the end of incubation; the presence influenza A virus can be confirmed by immunodiffusion or real-time reverse transcription polymerase chain reaction (real-time RT-PCR) in allantoic fluids [69,70,71].

#### 3.2.3. Antigen Detection

The presence of influenza virus type A can be identified by agar gel immunodiffusion (AGID) by checking for antigens in the nucleocapsid or cellular matrix, which is common in all subtypes of influenza type A [5,69].

Viral antigen detection can also be performed by enzyme immunoassay (EIA) and immunofluorescence assays that are generally sold commercially. These kits are based on immunochromatography (lateral flow devices) and use monoclonal antibodies against nucleoproteins [69,72]; however, they have been shown to have highly variable sensitivity, so they may not correctly identify infected animals [73].

#### 3.2.4. Antibody Detection

Antibody detection is performed using an enzyme-linked immunoadsorption assay (ELISA), which can detect antibodies to the cell nucleus capsid protein [73]. The assays may be carried out in both “competition” and “blocking” formats, detecting antibodies against AI viruses [37].

#### 3.2.5. Viral RNA Detection

Viral RNA detection is performed by reverse transcription polymerase chain reaction (RT-PCR) and real-time RT-PCR, which yield the characterization of influenza A, identifying subtypes H5 and H7 and its Eurasian and Pan-American lineages directly from infected samples [1,37,40].

All molecular diagnostic tests should be performed under strict precautions with specialized biosafety levels (BSL-2) [40,74].

## 4. Management of a Possible Avian Influenza Outbreak

### 4.1. Strategies to Deal with a Suspected Case

The suspected case may occur in a poultry farm, an animal slaughtering center, or in an isolated manner in wild or artisanal birds [10]. In any case, epidemiological surveillance protocols must be initiated by authorities until a definitive diagnosis is obtained [75].

It is suggested to standardize the containment protocol utilizing the following procedures [1,76]:Whenever trained personnel are available, carry out a clinical diagnosis and, if possible, a necropsy of the dead animals.Initiate an epidemiological investigation to determine possible forms of transmission and contact with other birds.Proceed immediately with sample collection and shipment to specialized laboratories.Create a register of farms, poultry, and domestic fowl within the suspected outbreak area.Create communication channels between the owners of the affected animals and the sanitary authorities so that they understand the situation until the diagnosis is confirmed.Report to international institutions, such as the World Organization for Animal Health (OIE) and the Food and Agriculture Organization of the United Nations (FAO), so they can take inter-institutional coordination measures.

### 4.2. Strategies for a Confirmed Case

When an AI case has been confirmed, containment actions should be prompt and strict precautions should be taken when handling infected poultry cases or infected materials. The protocol outlined below should be followed [1,40]:Generate poultry records of each nearby farm, including backyard poultry.Conduct an epidemiological investigation to identify the mode of transmission and possible contacts to prevent new cases.The epidemiological fence should be managed as follows (Figure 2):Infected Zone: It will be integrated by poultry farms and domestic breeding animals within a radius of 1 km around case 0; in this zone, the recommendations indicate drastic measures, such as the depopulation of flocks and sanitary culling in infected sectors. Additionally, the elimination of carcasses, products, and by-products of all poultry, executing cleaning and disinfection procedures of all infected material, should occur.Observation Zone: Poultry farms and domestic livestock within a radius of 3 km around case 0; in this zone, the transport of poultry products should be prohibited, and health sensors should monitor a possible outbreak.Surveillance Zone: Poultry farms and domestic livestock within a 7 to 10 km radius around case 0; a safety margin should be established by closing stores and markets selling poultry and eggs within a 10 km radius around the infected outbreak.

In addition, all entrances and exits to each zone should be thoroughly disinfected to avoid spreading the virus between zones [40].

## 5. Containment Measures for Confirmed Cases

### 5.1. Sanitary Slaughter of Infected Poultry

Within 24 to 48 h after confirmation of a positive case [1], all birds within a 3 km radius of the infected area should be monitored and traced to establishments with direct or indirect links to the infected premises. A tracing period should be considered, and AI high-risk establishment identification should be performed to opt for stamping out [40]. Sanitary slaughter could be an option for AI eradication, but it involves complex economic, ethical, environmental, and public health considerations due to the risk of zoonosis. However, if this procedure is chosen, animal welfare must be guaranteed [77] and be under the current legislation [78]. There are different procedures to be followed.

#### 5.1.1. Environment Saturation with CO_2_

Slaughtering by releasing carbon dioxide is one of the most recommended methods because large animal volumes can be euthanized with minimal contact by the responsible personnel. For example, in optimum conditions, a 50 L tank is enough to sacrifice 20–30 thousand birds [1]. This process can be carried out inside the same sheds where the animals are kept or by placing them in vans, cages, or airtight containers that ensure CO_2_ concentration remains stable [1]. Birds should first be exposed to concentrations below 40% CO_2_ and once they lose consciousness due to the anesthetic and central nervous system depressant gas effect, the concentration values can be raised to above 60% [79].

#### 5.1.2. Carbon Dioxide Foam

The animals must be placed at ground level, and the foam should be concentrated at 1% CO_2_ in water. The entire bird should be covered at approximately 1.50 m in height or 30 cm above their heads [1,80,81]. This is a very efficient method to deal with emerging outbreaks because total depopulation can be carried out quickly by inducing hypoxia in the animals [80,81].

#### 5.1.3. Cervical Dislocation

In case the previously mentioned methods are not available, euthanasia should be performed by manual or mechanical separation of the skull from the vertebral column after dislocating the neck of the bird [69]. Cervical dislocation is a non-invasive method; however, it requires trained personnel and is inefficient for large flocks or birds, in addition to the long exposure time of the responsible personnel [82].

### 5.2. Disposal

#### 5.2.1. Burial

For the disposal of the slaughtered birds and all the implements that have been in contact with the infected area (protective clothing, bedding, feed, and eggs), burial is a cost-effective and efficient option [1]. Heavy machinery and sufficient space should be available to dig the pits, occupying approximately 200–300 birds per cubic meter [1]. Long trenches that are not too large should be considered, and the channels should be covered with a layer of soil of about 40 cm, followed by an even layer of calcium dioxide and, finally, another layer of soil [40].

#### 5.2.2. Incineration

This method is not recommended for large poultry populations due to its high costs, environmental contamination, and lack of certainty in the effectiveness of the disposal method due to the volatility of the remains [1]. It is suggested to dig an 8 m long, 2 m wide, and 1 m deep trench; use firewood to completely burn the dead birds; and then add a layer of lime [83]. The channels should be covered with a layer of soil, a layer of lime, and finally, another layer of soil [40].

### 5.3. Infected Poultry Farms’ Disinfection

After slaughter and disposal of carcasses and other contaminated products, recommendations indicate that a three-step cleaning and disinfection protocol must be carried out [1,84]:The first step is to spray all surfaces that may have been in contact with the affected birds with cationic surfactants, oxidizing agents, aldehydes, or acids and leave them for one day [84].The second step is general cleaning with hot water or steam using degreasing and sulfating agents, finishing with disinfectant application for one week.The third step involves the same procedure as the second step but, optimally, leaving the disinfectant for 21 days [1].

## 6. Avian Influenza Control and Surveillance

The sanitary period should be at least four weeks within the infected zone. Once no new outbreaks have appeared in the observation and surveillance zone, trade can be resumed within the epidemiological fence. If vaccination plans are implemented, trade closure should be maintained.

aVaccination

Throughout history, vaccination has eradicated some diseases; however, most influenza vaccines target the HA. Therefore, vaccines must be continuously updated as HA mutates due to antigenic drift or reassortment [85]. The advantages of vaccination include rapid alleviation and the chance to eliminate the virus without interacting with wild bird reservoirs. On the other hand, vaccination can be costly, which could severely impact the economy of the farms and may promote diversification and viral mutation. Moreover, mass vaccination strategies have been successful in different countries. For instance, the H5 HPAI clade 7.2 has been almost eliminated from China due to mass vaccination [85,86].

In response to the increasing wave of HPAI cases, the Emergency Prevention System for Transboundary Animal and Plant Pests and Diseases (EMPRES) and the Terrestrial Animal Health Code have implemented general suggestions, which indicate that all efforts should focus primarily on the following [4,87,88,89,90,91]:bActive surveillance for early warning of HPAI in poultry.
Strengthening surveillance systems for timely identification of highly pathogenic AI and isolation of infected animals.Permanent monitoring of animals, health sensors, and the population that has contact with domestic poultry will allow the health authorities to carry out rapid AI infection detection.All suspected cases of highly pathogenic avian influenza should be reported to the animal health authorities for investigation, and samples should be taken and sent for laboratory analysis.A consensus should be reached between owners and authorities for periodic diagnostic testing for birds at high risk of infection.Increase biosecurity measures in poultry production and train personnel in good animal husbandry and manufacturing practices.Generate strategies for the identification of poultry workers, operators of poultry slaughter centers, and exposed persons within the food chain who should remain in isolation for approximately ten days from the last contact.
cPassive surveillance of wild birds
Surveillance of wild birds should be carried out according to the seasonality of the virus and the periods in which birds migrate to certain places, reinforcing surveillance in periods with more cases.When mortality in wild birds is observed, health authorities should be alerted and initiate the process of collection and testing for viral identification.When positive cases of HPAI are detected in a country or region, surveillance protocols for wild birds should be initiated, as the movement of migratory waterfowl is considered a potential risk for virus transmission into non-infected areas.


## 7. Discussion

Research has reported that regardless of the pathogenicity of the avian influenza strain (LPAI or AAP), the consequences on the economy of the regions are directly or indirectly affected [2]. Avian influenza has a high mortality; however, to avoid the propagation of the virus, even in asymptomatic birds, animals that have been in contact with the positive case must be euthanized. Moreover, animal welfare and operators’ health must be considered when carrying out sanitary depopulations [92].

The AI virus has wide antigenic variability and exponential mutation capacity, giving it a broad virulence spectrum and the ability to infect different species, including humans [23]. AI viruses can cross the interspecies barrier through a series of antigenic adaptations created by viral nucleoproteins in the cellular matrix; numerous scientific communities mention that the highest risk factor for public health is the continuous and periodic infection of mammalian hosts, which increases the possibility of virus adaptation [4,32,34].

A wide range of technologies is available for avian influenza diagnosis. According to CDC guidelines, rapid antigen detection tests, such as immunofluorescence or enzyme immunoassay, should not be the diagnostic method of choice in the event of a suspected outbreak of avian influenza [7,93] because they have low sensitivity, yielding a high number of false positives, and do not discriminate between viral subtypes, which has limited diagnostic utility [59,94]. However, if resources are limited, WOAH suggests the use of antibody detection assays, such as agar gel immunodiffusion (AGID), although it can only be interpreted as a collective result and not as an individual test [10]. Kanaujia and his collaborators agree with these recommendations and suggest that cell culture for viral amplification should not be the diagnostic method of choice due to the aggressiveness of HPAI viruses that rapidly kill the culture medium and the long time required to obtain the results for emergency situations [10,40,90]. For these reasons, the WHO, the CDC, and several scientists suggest the use of reverse transcriptase polymerase chain reaction (PCR) and real-time reverse transcriptase polymerase chain reaction (RT-PCR) as the principal diagnostic methods [44,81,95,96]. PCR is the chosen diagnostic method based on the ease of viral RNA extraction and amplification, the possibility of targeting the study for the subtypes of interest, and the speed with which the results are obtained, indicating when a case of influenza virus type A infection is suspected [40,73,89,97].

Moreover, an important vision that cannot be left aside is next-generation sequencing, which allows us to perform targeted sequencing and analysis of the complete viral genome, which would help us to understand the virus epidemiology according to the chronology and distribution of the outbreak [95,98,99].

During the intervention before an outbreak of avian influenza, the General Secretariat of Agriculture and Food and the Spanish Public Health Commission, the National Service of Health, and Agrifood Quality of Argentina, as well as other scientific collaborators, are in accordance that surveillance ranges must be implemented around the detected case with total restrictions of mobilization and immediate euthanasia intervention [1,39,75],[76,94]. Several studies have evaluated the efficacy of killing poultry by saturation of the environment with CO_2_, indicating that it is a valid method in case of a sanitary emergency in animals raised in superimposed pens [100]. However, this technique requires total and controlled isolation of the environment, and not all farms are suitable to apply gassing; if CO_2_ stunning is administered partially, with leaks inside the houses, the death of the birds will not occur in the desired way, causing them prolonged periods of asphyxia and suffering [101,102,103].

As a viable alternative, depopulation of birds using high-expansion CO_2_ foam is considered a slaughter method that reduces the exposure time and the number of operators necessary to perform it [96]. This method has demonstrated better physiological responses in the birds before their death due to its speed and easy applicability. A limitation is that this method is focused on birds raised on the floor [1,80,97,98,103]. Similarly, cervical dislocation is another method for euthanasia [69]. However, in the case of an outbreak with a high density of birds, this approach will be impractical [82]. Several studies show that this form of slaughter, with or without previous anesthesia, is detrimental to the welfare of the birds [104]. Moreover, this technique relies on the operator’s experience [105,106], leaving the animals susceptible to bad practices [107,108].

Once the euthanasia process has finished, the carcasses and infectious waste should be disposed within 48 h post-mortem [1,40,70]. Burial is a practical alternative when the necessary space is available based on the density of birds (between 150 to 300 birds per cubic meter according to the weight of the birds) [70,107,109]. Furthermore, heavy machinery is required to guarantee a pit depth of approximately 3 m so that the animals are at least 1 m below the natural surface [110]. It is important to emphasize that before applying this method, the phreatic mantles of the geographical area should be evaluated to avoid water contamination by runoff of fluids [1,111]. Incineration may be another method applicable in the field for small flocks [1]. However, it is not very accessible because it requires comburent and permanent surveillance of the fire quality for a period greater than or equal to 48 h [70,112]. Moreover, it generates environmental contamination and the possibility that the ashes and volatile residues contaminate surrounding sites [83,110].

After the elimination of the slaughtered animals, organic residues, and inert materials that have been near the infected birds, all areas of the affected premises should be cleaned and disinfected with activities that include detergents and disinfectants, alternated for a period of 21 to 30 days after the end of the elimination of infectious waste [1,71]. The WOAH, in a recent study, determined that the best way to eliminate the virus from inside the facilities and surfaces is dry cleaning and heat or steam applications [109]; however, due to the surface extension of poultry facilities, international and local guidelines suggest that cleaning with clean water and neutral detergents should be followed by antioxidant agents such as disinfectants. It is important to maintain the appropriate times for each period. This method is a viable alternative for virus elimination in poultry facilities where dry disinfection is not available [1,75,76,113,114].

Furthermore, there is a need to evaluate the pertinence of a vaccination plan as a measure of prevention and propagation of the disease. It should be considered that existing vaccines protect birds against infections; however, they require strict management and revaccination measures and may not inhibit viral shedding [1,71,75,76].

## 8. Conclusions

The goal of this work is to provide strategies to help healthcare professionals manage an outbreak and understand the impact of avian influenza. The recommendations proposed in this article are a compilation of different international guidelines that have been adopted in some countries and may be useful for developing policies or regulations in the event of an avian influenza outbreak. Avian influenza is a disease that varies greatly depending on the region and the species it infects, so general recommendations are discussed. The guidelines presented in this document should be considered according to the contact frequency between domestic and wild species, as well as the level of biosecurity managed in each region and the susceptibility or protection of the avian species.

## 9. Future Directions

The present work describes strategies to manage an avian influenza outbreak; however, further efforts must be made to develop strategies for outbreak prevention. Moreover, other strategies could include improving hygienic practices, vaccination plans, and diagnostic methods for early detection and control.

## Figures and Tables

**Figure 1 pathogens-12-00610-f001:**
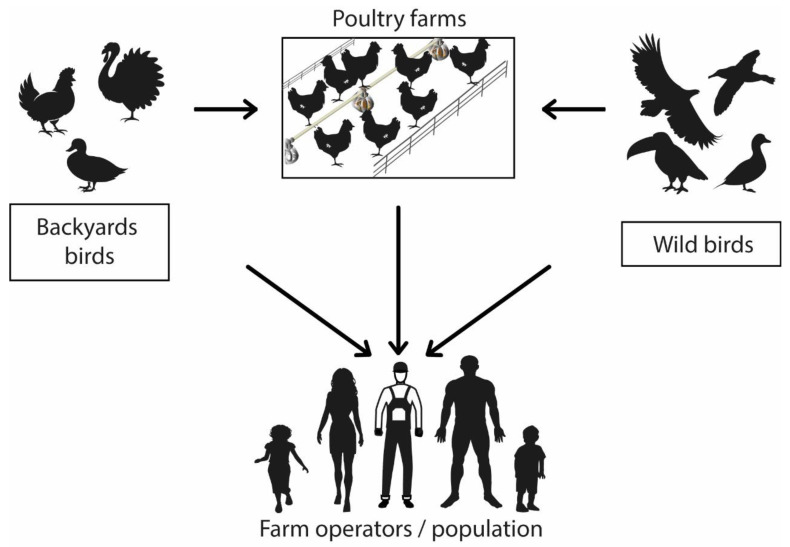
Avian influenza virus transmission mechanism. Graphic representation of virus zoonotic potential.

**Figure 2 pathogens-12-00610-f002:**
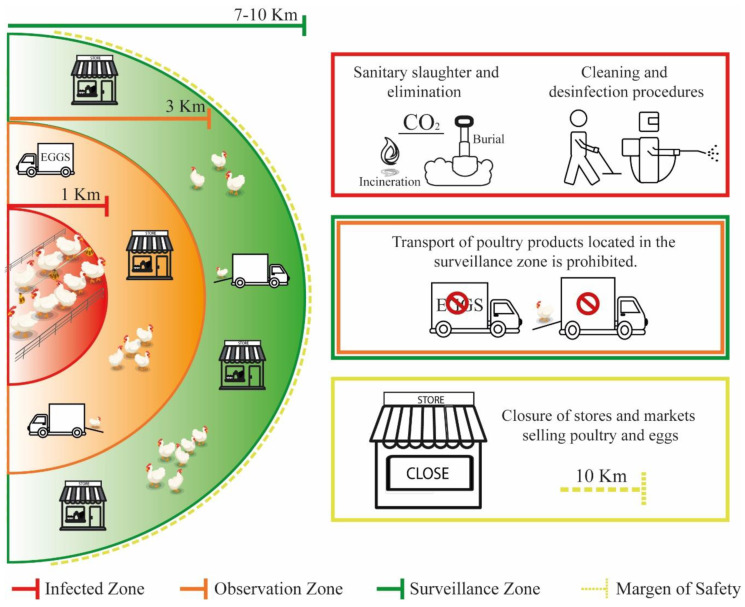
Epidemiological fence for the management of avian–influenza–infected areas.

## Data Availability

The results are presented in the paper. For more information, please contact the corresponding authors.

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
