# Peer review of "Avian Influenza: Strategies to Manage an Outbreak"

_pathogens, 2023, doi:10.3390/pathogens12040610_

Round 1

Reviewer 1 Report

 Simancas-Racines et al. reported avian influenza with particular attention to the strategies to manage outbreaks. The review is very interesting and well-organized. However,  I recommend adding a chapter about vaccination. 

Other minor revision:

- Family and orders should be in italics (Orthomyxoviridae (line 94), Anseriformes, and Charadriiformes (line 187)).

- Line 119:  Low pathogenic avian influenza virus should be LPAI

-. Line 132: highly pathogenic influenza should HPAI 

- Line 98: Please describe the virus. " The segmented nature of the influenza virus genome leads to emerging of new viruses through genetic reassortment, making its prevention and control more difficult (Parvin et al. 2022; Doi: https://doi.org/10.51585/gjm.2022.3.0016). 

Reviewer 2 Report

Avian influenza virus (AIV) is a type of IAV that is isolated from and adapted to avian host species, and it is classified as low or highly pathogenic. Highly pathogenic avian influenza viruses continue to pose a huge threat to global animal and human health. This review is nicely written and the structure of the document is logical, I have only minor comments that authors could consider.

Specific comments

·         In the abstract is mentioned that “Moreover, AI is a notifiable animal disease; hence all countries must report infections by LPAI and HPAI to the health authorities.”. However, only HPAI (or H5 and H7 LPAI) are notifiable disease. Please clarify this point.

·         Section 2.1. Etiology: Include influenza D virus.

·         Section 2.4.2. Zoonotic transmission, lines 171-172 “Certain mammals can act as reservoirs”. Please, mention which ones or at least a few examples as pigs….

·         Sometimes it is unclear if the proposes measures are recommendation for international organizations (WOAH, FAO, others…). Measures already established in some countries….Please clarify.

·         Lines 413-414 “AI viruses manage to cross the interspecies barrier through a series of antigenic adaptations generated by viral nucleoproteins in the cellular matrix “. This sentence is unclear, please consider rephrasing.

·         It can be great if authors include a section regarding vaccination, advantages/disadvantages for vaccination policies, eg difficulties for trade after vaccination, surveillance tasks, etc….
